# *BRAF*$^{V600E}$-mutated serrated colorectal neoplasia drives transcriptional activation of cholesterol metabolism

Paulina Rzasa [1,8], Sarah Whelan[1,8], Pooyeh Farahmand [1,8], Hong Cai[1], Inna Guterman[1], Raquel Palacios-Gallego[1], Shanthi S. Undru[1], Lauren Sandford [1,7], Caleb Green [1], Catherine Andreadi[1], Maria Mintseva[1,2], Emma Parrott[1], Hong Jin[1], Fiona Hey[1], Susan Giblett[1], Nicolas B. Sylvius[3], Natalie S. Allcock[4], Anna Straatman-Iwanowska[4], Roberto Feuda[5], Cristina Tufarelli [1], Karen Brown[1], Catrin Pritchard[1] & Alessandro Rufini [1,6 ✉]

*BRAF* mutations occur early in serrated colorectal cancers, but their long-term influence on tissue homeostasis is poorly characterized. We investigated the impact of short-term (3 days) and long-term (6 months) expression of *Braf*$^{V600E}$ in the intestinal tissue of an inducible mouse model. We show that *Braf*$^{V600E}$ perturbs the homeostasis of intestinal epithelial cells, with impaired differentiation of enterocytes emerging after prolonged expression of the oncogene. Moreover, *Braf*$^{V600E}$ leads to a persistent transcriptional reprogramming with enrichment of numerous gene signatures indicative of proliferation and tumorigenesis, and signatures suggestive of metabolic rewiring. We focused on the top-ranking cholesterol biosynthesis signature and confirmed its increased expression in human serrated lesions. Functionally, the cholesterol lowering drug atorvastatin prevents the establishment of intestinal crypt hyperplasia in *Braf*$^{V600E}$-mutant mice. Overall, our work unveils the long-term impact of *Braf*$^{V600E}$ expression in intestinal tissue and suggests that colorectal cancers with mutations in *BRAF* might be prevented by statins.

[1] Leicester Cancer Research Centre, University of Leicester, Leicester, UK. [2] Area of Neuroscience, International School for Advanced Studies (SISSA), Trieste, Italy. [3] NUCLEUS Genomics, Core Biotechnology Services, University of Leicester, Leicester, UK. [4] University of Leicester Core Biotechnology Services Electron Microscopy Facility, Leicester, UK. [5] Department of Genetics and Genome Biology, University of Leicester, Leicester, UK. [6] Dipartimento di Bioscienze, University of Milan, Milan, Italy. [7] Present address: Institute of Cancer and Genomic Sciences, University of Birmingham, Birmingham, UK. [8] These authors contributed equally: Paulina Rzasa, Sarah Whelan, Pooyeh Farahmand. ✉email: Alessandro.Rufini@unimi.it

The single-layered intestinal epithelium contains functionally distinct cellular populations. Proliferating intestinal stem cells (ISCs), whose self-renewal depends on active Wnt/β-catenin signaling[1], are interspersed among Paneth cells at the bottom of epithelial invaginations, known as crypts of Lieberkühn, and can be identified by selective markers such as LGR5 and OLFM4[2,3]. ISCs generate transient amplifying cells, which have limited proliferative capability and migrate upwards towards intestinal villi where they differentiate into the absorptive enterocyte lineage or the secretory lineage, which includes Paneth cells, enteroendocrine cells (EECs), and goblet cells.

Colorectal cancer (CRC) is a heterogeneous disease arising from the intestinal epithelium through two main routes[4,5]. First, tumors originating from tubular, villous or tubulovillous precursor adenomas localize to the left colon, are characterized by sustained Wnt/β-catenin signaling, and are thought to develop from ISCs through a bottom-up fashion[6–9]. Second, tumors located in the right colon arise from precursor sessile serrated lesions, characterized by a saw-tooth-shaped folding of the dysplastic epithelium, a flat sessile morphology, and a mucinous histology[10]. This subtype of CRC develops in a top-bottom fashion through transdifferentiation of committed epithelial cells to gastric metaplasia and enrichment in fetal markers[6,7,11]. The most common driver mutation of serrated lesions is an activating mutation of the *BRAF* oncogene[12–14]. A valine to glutamate substitution at codon 600 (V600E) generates a constitutive active mutant (BRAF^V600E) serine-threonine kinase that is responsible for the activation of the downstream MEK/ERK arm of the MAPK pathway[15]. Additional molecular features associated with right-sided colonic neoplasia include CpG island methylator phenotype (CIMP), microsatellite instability (MSI), and loss of TGF signaling[10–12,14,16]. Recently, heterogeneity of *BRAF*-driven CRCs has been uncovered through bulk RNA sequencing, and two main BRAF^V600E CRC subtypes have been identified based on gene expression profile[17]. The BM1 subtype displays a gene expression profile enriched in EMT-related processes, *KRAS* signaling, and immune response, whereas the BM2 subtype is enriched in cell-cycle and cycle checkpoints-related processes, such as target genes of the E2F transcription factors and genes involved in the G2 to M transition of the cell cycle.

*Braf*^V600E has been reported to be a poor oncogene when expressed in the mouse intestinal epithelium, whereby few tumors develop after long latency[18–20]. This prolonged latency agrees with the knowledge that human CRC develops covertly over 10–15 years[21] and indicates that intestinal tissue harbors the oncogenic *BRAF* mutation for a considerable time despite the absence of any clinical manifestation of the disease. Notwithstanding, little is known about the long-term impact of *BRAF*^V600E expression on intestinal homeostasis, and the mechanisms that enable *BRAF*-driven CRC development remain incompletely understood. Recently, the limited intestinal tumorigenesis of *Braf*^V600E mutant mice has been attributed to an imbalance between stemness and differentiation, whereby expression of *Braf*^V600E promotes differentiation of intestinal epithelial cells at the expense of ISCs[18,22].

The overall aim of the work described here was to attain a deeper understanding of serrated tumorigenesis and ascertain the impact of long-term expression of *Braf*^V600E. To this end, we used a mouse model carrying an inducible *Braf*^V600E allele[20] targeted to the intestinal epithelium and performed transcriptional profiling over a 6-month time course. Together with dynamic changes in tissue homeostasis, we show that *Braf*^V600E orchestrates a rapid and persistent transcriptional reprogramming characterized by the enrichment of numerous gene signatures associated with *Braf*-driven CRC and additional signatures suggestive of metabolic rewiring. In particular, we observed a robust increase in the

expression of cholesterol biosynthesis genes. The functional relevance of those adaptations was confirmed by the ability of atorvastatin, a commonly prescribed pharmacological inhibitor of the mevalonate pathway and cholesterol biosynthesis, to prevent the establishment of crypt hyperplasia in *Braf*^V600E-mutant mice. Finally, through the analysis of human bulk and single-cell transcriptomic datasets, we confirmed that an increased cholesterol gene signature is a hallmark of serrated CRC.

## Results

**Activation of *Braf*^V600E in the intestine leads to crypt hyperplasia and persistence of crypt-resident ISCs.** Mice carrying an inducible *Braf*^V600E knock-in allele were crossed with *VillinCre*^ER mice to generate *VillinCre*^ER/0/*Braf*^V600E-LSL/+ (BVE) and control *VillinCre*^ER/*Braf*^+/+ (WT) mice. Intraperitoneal injection of tamoxifen in BVE mice enabled expression of the mutant *Braf*^V600E allele. Mutant BVE mice displayed reduced survival (median survival 414 days versus 622 days in WT mice) (Supplementary Fig. 1a) and, as previously reported, neoplasia developed after long latency and with limited penetrance (Supplementary Fig. 1)[18–20]. To gain insight into short- and long-term outcomes of oncogene expression, we performed tissue histology (3 days, 6 weeks and 6 months after induction of *Braf*^V600E) and transcriptomic analysis (3 days and 6 months after induction) of intestinal tissue of BVE and control mice (Fig. 1a and Supplementary Fig. 2). Analysis of transcriptomic data using GSEA confirmed a robust expression of a MAPK gene signature in mutant mice (Supplementary Fig. 3).

First, we assessed whether expression of mutant *BRAF* alters homeostasis of the intestinal crypt. Cell count indicated that as early as 3 days post induction, mutant crypts were hyperplastic (Fig. 1b and Supplementary Data 1). Hyperplasia was linked to increased cell proliferation (Fig. 1c, Supplementary Data 1) and was persistent, with a significantly increased number of cells detected in intestinal crypts of aged mutant mice (Fig. 1d and Supplementary Data 1). Previous reports have suggested that the expression of *Braf*^V600E causes transient amplifying cell generation and cellular differentiation at the expense of the ISC population[18,22]. Using the stem cells marker Olfm4[2], we were able to identify a persistent Olfm4+ population localized at the bottom of the intestinal crypt in tissue specimens collected 3 days, 6 weeks, and 6 months after induction (Fig. 1e). Moreover, stem cell marker genes *Lgr5* and *Olfm4* in BVE mice were expressed at levels comparable to those in control mice 6 months after *Braf*^V600E expression (Supplementary Fig. 4a). We also interrogated transcriptomic data, performing GSEA of an ISC signature. This analysis showed a mild downregulation early after *Braf* activation, but this was reverted in 6-month tissues (Supplementary Fig. 4b).

However, whereas in WT mice ISCs were interspersed with Paneth cells, in long-term induced mice ISCs clustered at the bottom of the intestinal crypt in long-term induced mice (Fig. 1e, insets), following a generalized loss and delocalization of Paneth cells (Supplementary Fig. 5). Finally, in keeping with the role of *Braf*^V600E in serrated neoplasia, villi from BVE mice developed a saw-toothed appearance early after oncogene expression (Supplementary Fig. 6a) and, consistent with the establishment of tissue hyperplasia, villi length was significantly increased in 6-week and 6-month mutant mice (Supplementary Fig. 6b).

**Expression of *Braf*^V600E modifies the abundance and distribution of differentiated intestinal populations.** We then assessed the impact of mutant *Braf* on differentiated intestinal cell populations. As early as 6 weeks post induction of *Braf*^V600E, the intestinal crypts were deprived of Paneth cells, which migrated

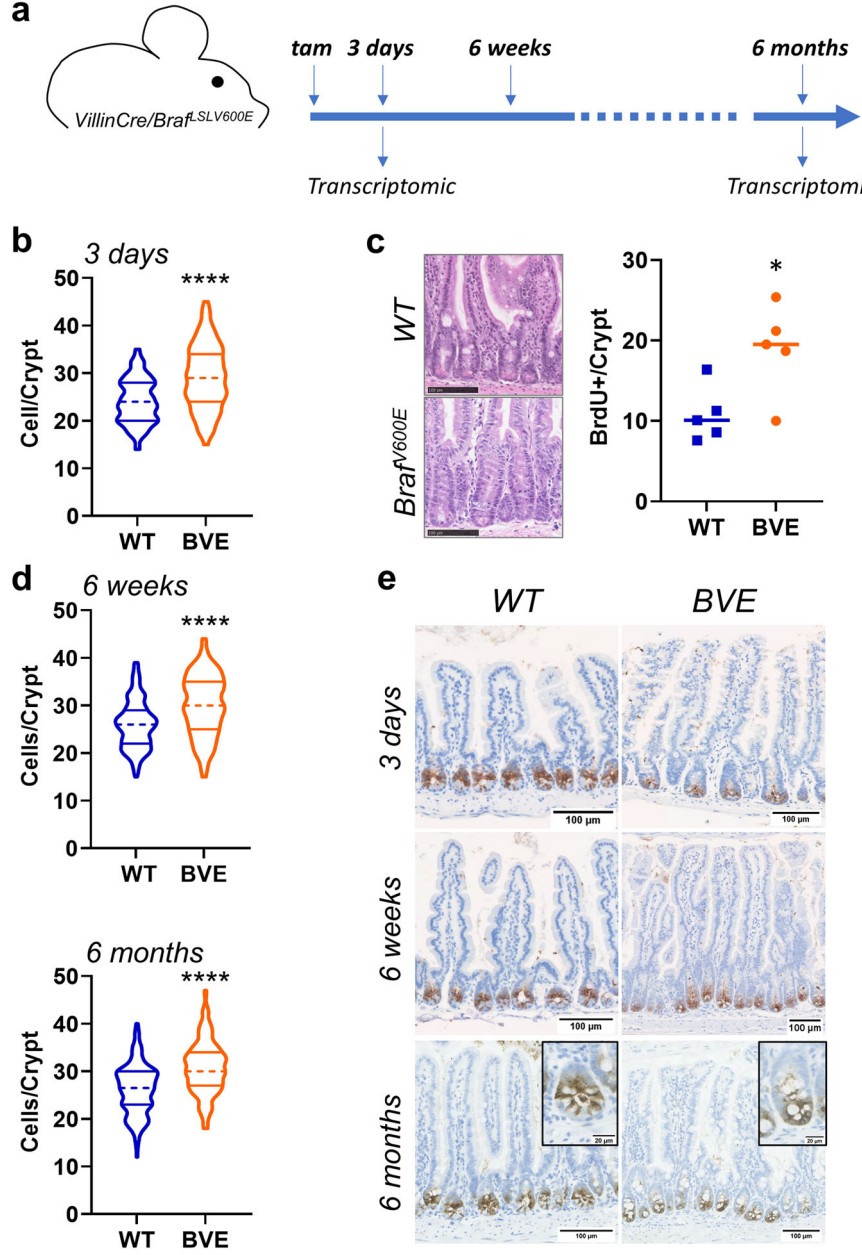

**Fig. 1 Persistence of ISCs and crypt hyperplasia in *Braf*<sup>V600E</sup> mouse intestinal tissue. a** Schematic of the experimental work: *VillinCre*$^{ER/0}$/*Braf*$^{V600E-LSL/+}$ were culled 3 days and 6 months post induction of *Braf*$^{V600E}$ for tissue histology and transcriptomic analysis. Mice were also culled at an intermediate time point 6 weeks post induction to corroborate changes in tissue histology. **b** Violin plots showing the distribution of number of cells per crypt in control mice and mice induced with *Braf*$^{V600E}$ for 3 days (BVE). Dotted and solid lines indicate median and quartiles, respectively. Data were analyzed by Kolmogorov–Smirnov test. $n = 6$ mice per group in 3 days, 4 WT and 3 mutant mice in 6 weeks, $n = 4$ mice per group in 6 months. 50 crypts were counted per each animal. ****$P \le 0.0001$. **c** H&E representative images of hyperplastic crypts and quantification of proliferative cells (BrdU + ) 3 days post tamoxifen injection ($n = 5$ animals per group, 1 female and 4 males, 31 crypts were counted for each animal, see also Supplementary Data 1). Data were analyzed using unpaired two-tailed *t* test. *$P \le 0.05$. Bar size = 100 μm. **d** Violin plots showing the distribution of number of cells per crypt in mice induced for 6 weeks ($n = 4$ control males and 3 mutant males) and 6 months ($n = 4$ mice per group). Dotted and solid lines indicate median and quartiles, respectively. Data were analyzed as in (**a**). ****$P \le 0.0001$. **e** Histological images of small intestinal tissue stained with the stem cell marker Olfm4 from WT control mice and mice with the *Braf*$^{V600E}$ mutation (BVE) at 3 days, 6 weeks and 6 months following tamoxifen induction. Bar size = 100 μm.

upwards toward the villi (Supplementary Fig. 5) and were progressively lost, a phenotype previously observed in mouse models with altered MAPK activity[23].

In addition, GSEA revealed an initial increase in the expression of an intestinal differentiation gene signature 3 days post *Braf*$^{V600E}$ induction. However, unexpectedly, the same signature was downregulated following long-term expression of mutant *Braf* (Fig. 2a). At this stage, we also identified a significant

downregulation of an intestinal enterocyte signature, which was reflected by reduced Alpl histological staining (Fig. 2b, c). A late impairment in the differentiation of enterocytes was confirmed by TEM, which showed shortened and less abundant microvilli on the surface of mutant enterocytes (Fig. 2d).

Next, we visualized and quantified mucus-producing goblet cells and EECs using Alcian Blue staining and immunofluorescence staining for Chromogranin A (ChgA), respectively (Supplementary

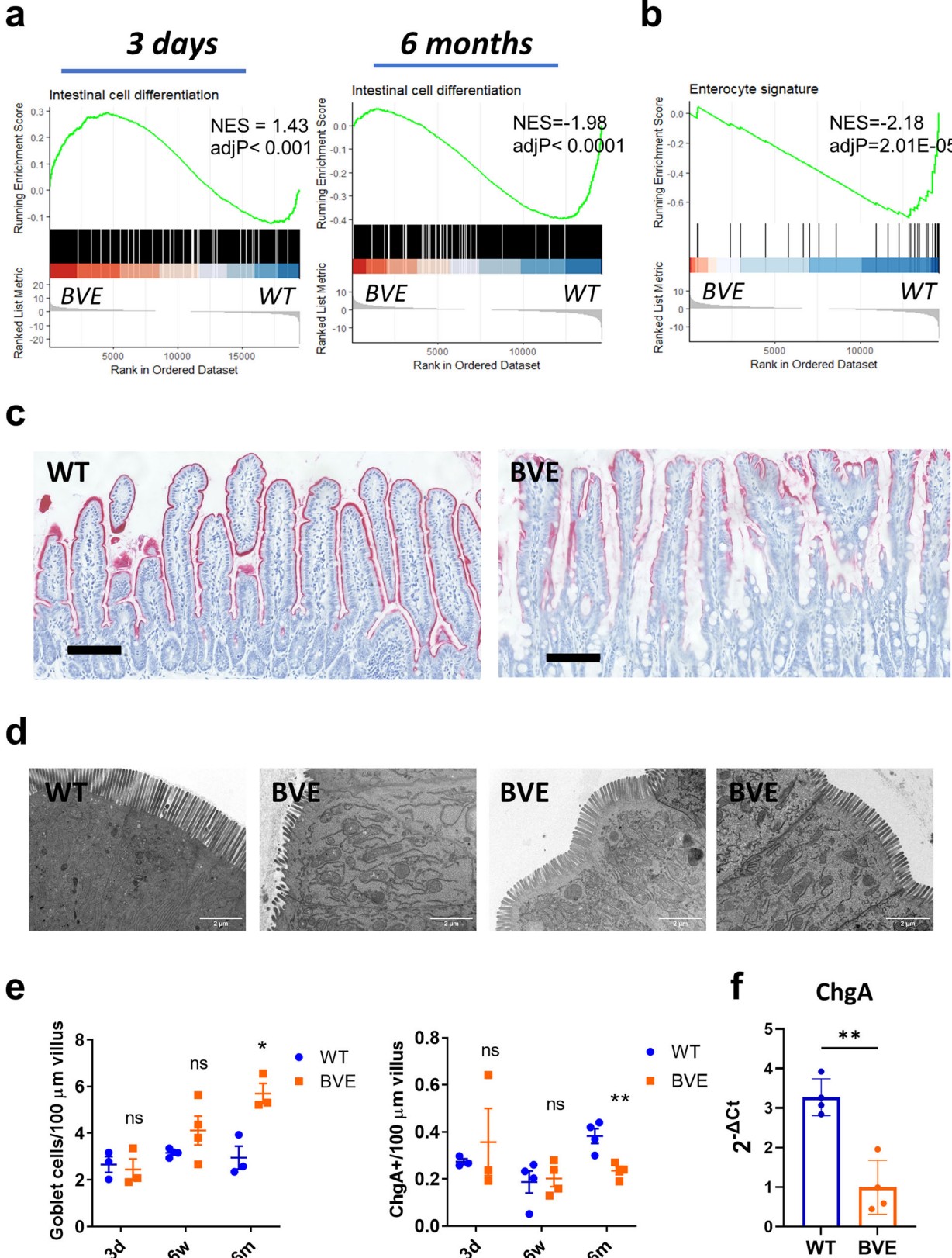

Fig. 7). Quantification of Alcian Blue- and ChgA-positive cells revealed an increased number of goblet cells, whereas EECs were reduced after long-term (6 months) expression of *Braf^V600E* (Fig. 2e and Supplementary Data 1). rt-qPCR analysis confirmed the reduced expression of ChgA in aged mice (Fig. 2f and Supplementary Data 1).

These data suggest that *Braf^V600E* alters the balance of intestinal secretory and absorptive cell lineages. The analysis of the long-term impact of oncogene expression particularly unveiled a significant impact on the numbers of goblet cells and EECs, and a broader downregulation of gene signatures

**Fig. 2 *Braf*[V600E] expression in the mouse intestinal epithelium alters intestinal homeostasis and affects absorptive and secretory cell lines. a** GSEA showing changes in differentiation gene signatures 3 days and 6 months following tamoxifen induction. **b** GSEA showing changes in enterocyte gene signatures 6 months following tamoxifen induction. NES normalized enrichment score. **c** Histological images of small intestinal tissue from WT control mice and mice with the *Braf*[V600E] mutation (BVE) 6 months following tamoxifen induction stained with alkaline phosphate (Alpl) to assess enterocyte cells. Bar size = 100 μm. **d** TEM images of WT and *Braf*-mutant intestinal tissues showing altered microvilli in mutant mice. scale bar = 2 μm. **e** Quantification of Goblet and ChgA+ enteroendocrine cells from the indicated mice. Each dot represents a single mouse, and lines represent mean ± SEM. Data were analyzed by unpaired two-tailed *t* test (n = 3 per group). *$P \leq 0.05$, **$P \leq 0.001$. **f** rt-qPCR analysis of the expression of the intestinal biomarkers Chromogranin A, (*ChgA*, EECs) 6 months after induction of *Braf*[V600E]. Data are plotted as mean ± SD. Dots represent single replicates. Data were analyzed by unpaired two-tailed *t* test (n = 4 per group). *$P \leq 0.05$.

associated with intestinal differentiation and, more specifically, with enterocyte differentiation.

Overall, our data confirm previous evidence of reduced stemness and increased differentiation early after induction of *Braf*[V600E]. They show, in addition, that those changes are transient and that stemness is restored in the intestinal tissue at the expense of differentiation. These findings emphasize the importance of long-term analysis of oncogene expression in the intestinal tissue and they are congruent with recent data suggesting a top-down origin of human serrated neoplasia triggered by dedifferentiation of intestinal cells[6].

***Braf*[V600E] expression regulates a transcriptional program linked to CRC.** Next, we identified the molecular pathways regulated by *Braf*[V600E] by interrogating the Molecular Signatures Database (MSigDB)[24] (Supplementary Fig. 8). Signatures that scored high included pathways associated with cellular proliferation and the BM2 subtype of human CRC, such as Myc, E2F targets and G2/M checkpoint signatures (Fig. 3a), which were significantly enriched as early as 3 days post *Braf*[V600E] induction. The same signatures were confirmed to be enriched in *BRAF*[V600E] mutant CRCs compared with normal colon tissue from the TCGA dataset[8] (Supplementary Fig. 9). We also observed an enrichment of a canonical Wnt pathway signature in the mouse mutant intestinal tissue (Fig. 3b), in agreement with previous findings indicating that activation of *Braf*[V600E] results primarily in intestinal tumors displaying Wnt pathway activation[19,20,22]. More tellingly, long-term activation of *Braf*[V600E] was associated with the expression of *BRAF*-mutant CRC signatures that predict poor prognosis in CRC patients[25] (Fig. 3c).

To ascertain whether MAPK activity was necessary for the establishment of the observed transcriptional changes, we treated mice with the MEK inhibitor PD184352 (MEKi). Mice received daily treatments with vehicle or MEKi for 3 consecutive days and were then culled for transcriptomic analysis (Supplementary Fig. 10a). MEKi administration achieved a marked decrease in phospho-Erk in intestinal tissue, as assessed by Western blot analysis (Supplementary Fig. 10b), and stark repression of the MAPK signature[19] (Supplementary Fig. 10f). Notably, pharmacological inhibition of the MAPK pathway reverted the enrichment of some, but not all, signatures. In particular, the Myc signature, the canonical Wnt signature, and the Popovici poor-prognosis signature all depended on a functional MAPK pathway (Fig. 3d).

The origin of *BRAF*[V600E] CRC remains elusive. Recent evidence suggests that serrated neoplasia arises through intermediate gastric metaplasia and a reversion to an embryonic cellular stage, sustained by a fetal gene signature[6,7,11]. Indeed, human serrated lesions show enrichment for gene expression associated with metaplasia and fetal genes (Supplementary Fig. 11a). In agreement with previous observations[11], expression of *Braf*[V600E] triggers significant and persistent enrichment of the fetal signature in intestinal tissue, which we show to be dependent on MAPK signaling (Supplementary Fig. 11b). However, we did not detect significant changes in the metaplasia gene signature[6], although two genes within the signature, namely Aquaporin 5

(*Aqp5*) and Annexin A10 (*Anxa10*), were consistently upregulated in mutant tissue in a MAPK-dependent fashion (Supplementary Fig. 12). Interestingly, *Anxa10* has been previously reported to be a specific marker for human serrated lesions and CRCs of the serrated pathway[22,26,27].

Overall, these data indicate that activation of *Braf*[V600E] orchestrates changes in the intestinal transcriptome that reflect the transcriptional reprogramming underlying colorectal carcinogenesis. These changes are durable, persisting for up to 6 months following oncogene expression.

***Braf*[V600E] expression rewires gene signatures of cholesterol metabolism to drive crypt hyperplasia.** We noticed that the top-scoring gene signature enriched in *Braf*-mutant mice was a cholesterol signature at both 3 days and 6 months following *Braf*[V600E] induction (Fig. 4a). The robust increase in this signature was also MAPK-dependent (Fig. 4b). The leading-edge genes responsible for the signature enrichment included several key metabolic enzymes of the mevalonate pathway, such as *Idi1*, *Mvk*, *Fdft1*, *Fdps*, *Sqle*, *Hmgcs1*, *Mvd*, *Hmgcs2*. Increased expression of cholesterol biosynthesis genes was also confirmed by rt-qPCR on RNA extracted from intestinal tissue (Fig. 4c and Supplementary Data 1). Next, we confirmed enrichment of the cholesterol biosynthesis signatures in RNA-seq datasets from two independent mouse models of *BRAF*-driven CRC (Fig. 4d)[11,18].

Cholesterol biosynthesis contributes to tumorigenesis and can be inhibited using statins, a widely prescribed category of drugs that target the mevalonate pathway rate-limiting enzyme HMG-CoA reductase. Several studies indicate that regular statin use reduces the incidence of several malignancies, including CRC[28,29]. Hence, we reasoned that cholesterol biosynthesis contributes to the establishment of the durable crypt hyperplasia observed following induction of *Braf*[V600E]and its inhibition by statins would prevent the formation of hyperplastic crypts. To test this hypothesis, we treated mice with daily doses of atorvastatin starting 1 week before induction of *Braf*[V600E] and collected tissue 3 days post tamoxifen administration. Counting of crypt epithelial cells confirmed that statin treatment prevented the establishment of crypt hyperplasia in mutant intestinal tissue (Fig. 4e and Supplementary Data 1). Mechanistically, atorvastatin did not alter crypt proliferation (Fig. 4f and Supplementary Data 1), but significantly increased apoptosis assessed through cleaved PARP immunohistochemistry (Fig. 4g and Supplementary Data 2).

Overall, these data indicate that *Braf*[V600E] elicits a transcriptional reprogramming of cellular metabolism indicative of increased cholesterol biosynthesis, which contributes to the establishment of tissue hyperplasia by increasing the survival of crypt cells. The results suggest that statins might prevent CRC harboring mutant *BRAF*[V600E].

**Expression of *BRAF*[V600E] establishes a network of transcription factors that mediates the enrichment in the cholesterol biosynthesis gene signature in human serrated lesions.** Next, we investigated whether the enrichment of the cholesterol

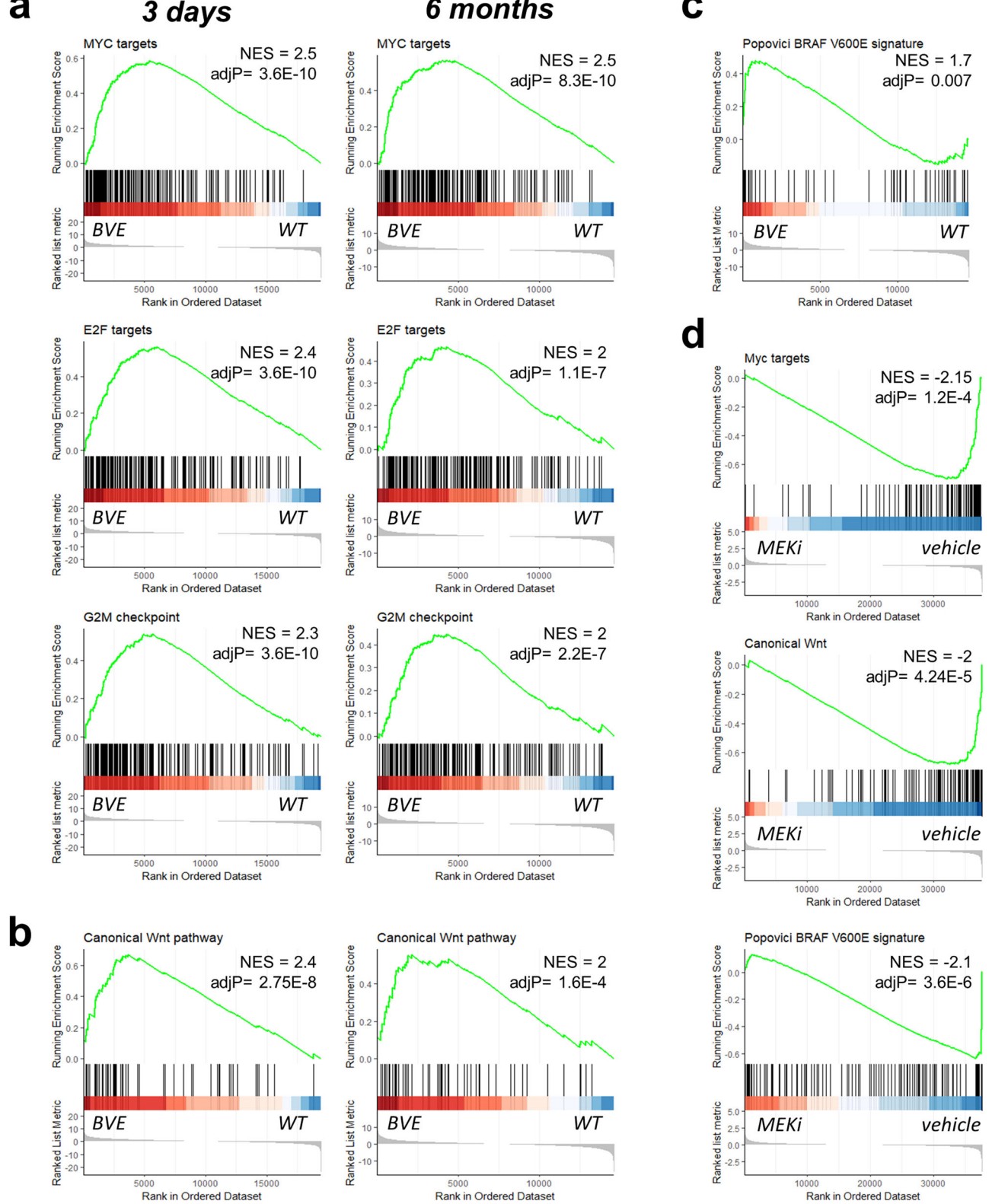

**Fig. 3 Expression of *Braf^V600E* in the intestinal epithelium induces persistent gene signatures related to CRC. a** GSEA showing changes in signatures associated with *BRAF^V600E*-driven CRC 3 days and 6 months following tamoxifen induction. **b** GSEA showing changes in the Wnt signature 3 days and 6 months following tamoxifen induction. **c** GSEA showing enrichment in a poor-prognosis gene signature for *BRAF^V600E*-drive CRC 6 months following tamoxifen induction. **d** GSEA showing dependence on MAPK pathway for the establishment of key gene signatures. Mice expressing *Braf^V600E* for 3 days were treated with vehicle control or the MEK inhibitor PD184352 (MEKi). NES normalized enrichment score.

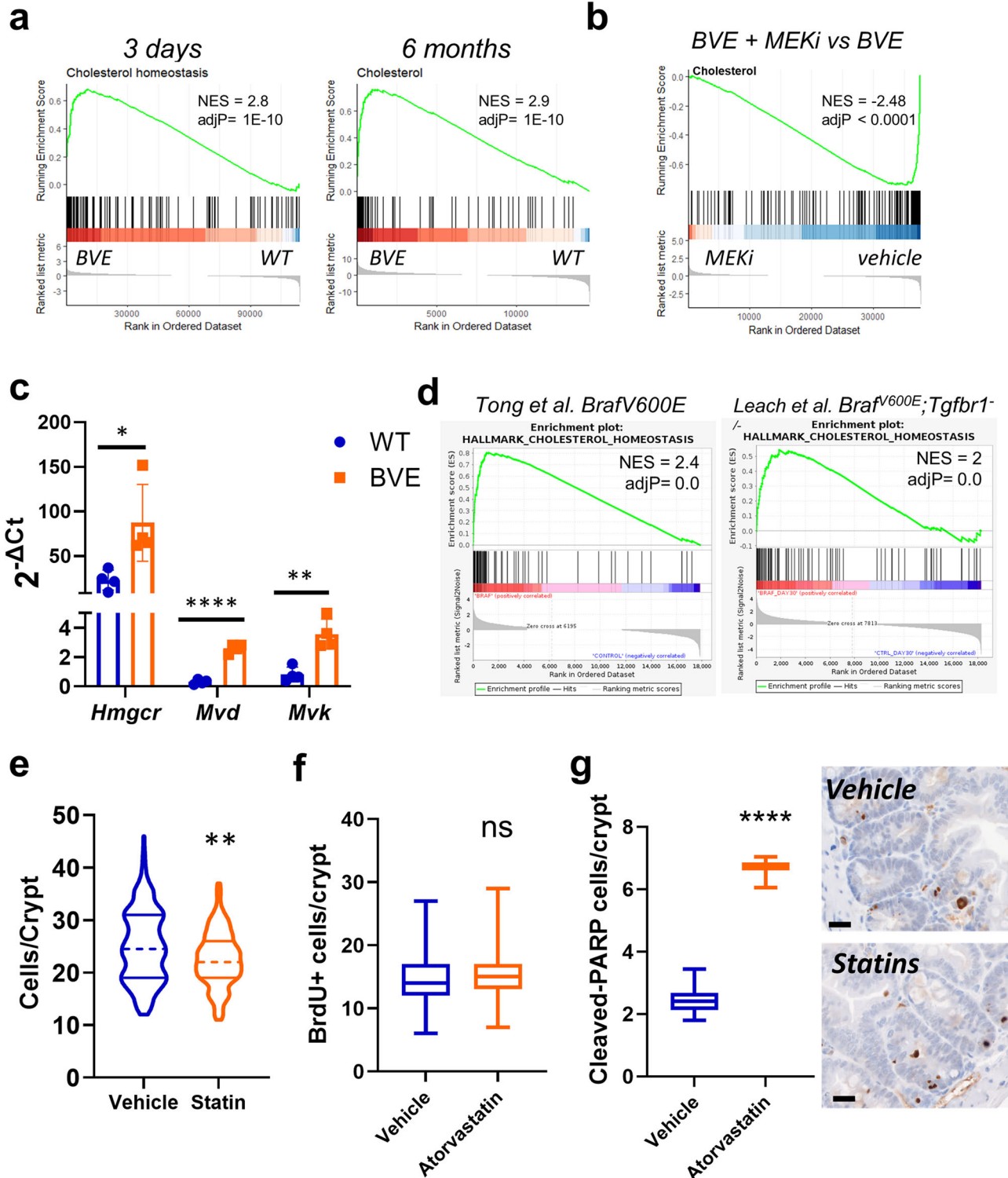

metabolism gene signature induced by $Braf^{V600E}$ in the mouse intestine was preserved in human serrated lesions. To this end, we analyzed three independent datasets of serrated lesions, which enabled comparison between serrated lesions, traditional adenomas and normal right colon tissue[30–32]. These analyses showed a robust enrichment in the cholesterol biosynthesis signature when serrated lesions were compared to either normal tissue or adenomas (Fig. 5a, b). Moreover, we confirmed in two independent datasets that CRCs harboring mutant $BRAF^{V600E}$ display an enrichment of the cholesterol gene signature compared to WT

cancers (Supplementary Fig. 13a)[7]. Notably, no significant enrichment was observed when adenomas were compared to normal tissue control, suggesting that the transcriptional rewiring of cholesterol metabolism is specific to serrated lesions (Supplementary Fig. 13b). Since bulk transcriptomic data cannot distinguish between the contributions by tumor or non-tumor cells to the enrichment of gene signatures, we ascertained expression of the cholesterol metabolism signature in transformed epithelial cells, by querying a scRNA-seq dataset of serrated lesions and normal tissue[6]. Clustering of the serrated lesions and normal

**Fig. 4 Braf$^{V600E}$ in the intestinal epithelium shows the enriched expression of a gene signature related to cholesterol metabolism. a** GSEA showing changes in gene expression of the cholesterol biosynthesis signature 3 days and 6 months following tamoxifen induction. NES normalized enrichment score. **b** Treatment of mice with the MEK inhibitor PD184352 (MEKi) reverts the enrichment of the cholesterol biosynthesis gene signature in intestinal tissue. **c** rt-qPCR analysis of cholesterol metabolism genes in intestinal tissue collected 6 months after tamoxifen injection. Bars represent mean ± SD. Dots represent single replicates. Data were analyzed by unpaired two-tailed $t$ test ($n = 4$ per group). **$P \leq 0.01$; ***$P \leq 0.001$. **d** GSEA showing increased transcriptional expression of cholesterol biosynthesis gene signature in the intestinal tissue of two independent Braf$^{V600E}$ mouse datasets. NES normalized enrichment score. **e** Violin plot showing the distribution of the number of cells per crypt in Braf-mutant mice induced for 3 days and treated with vehicle control or atorvastatin 10 mg/mL. Dotted and plain lines indicate median and quartiles, respectively. Data were analyzed by Kolmogorov–Smirnov test. $n = 6$ female mice in the vehicle group and 3 female mice in the atorvastatin-treated group. In total, 50 crypts were counted per each animal. **$P \leq 0.01$. **f**, **g** Cell proliferation and apoptosis were evaluated in Braf-mutant intestinal crypts of mice treated with vehicle control or atorvastatin using IHC for BrdU and cleaved-PARP1, respectively. Examples of cleaved PARP positive crypt cells are provided (bar size = 20 μm). Data are plotted as box and whisker plots. Boxes represent median and quartile values, whiskers represent min and max values. Data were analyzed using unpaired, two-tailed $t$ test ($n = 6$ vehicle and 3 atorvastatin-treated female animals). ns not significant, ****$P \leq 0.0001$.

single cells revealed seven canonical cell types and one serrated specific cell (SSC) subtype[6] (Fig. 5c and Supplementary Fig. 14a). Notably when the expression of the cholesterol biosynthesis signatures was scored in the cell clusters using the UCell tool[33], SSCs exhibited significantly higher expression of the signature, compared with other cell types (Fig. 5d, e and Supplementary Data 1), including cancer cells from conventional adenomas (Supplementary Fig. 14b). Key metabolic enzymes of the mevalonate pathway were expressed in SSCs, some of them showing a significant enrichment compared to non-cancerous cells (Supplementary Fig. 14c).

These data indicate that the transcriptional metabolic rewiring of cholesterol metabolism observed in Braf$^{V600E}$ mouse intestinal tissue recapitulates comparable adaptations of human colorectal lesions and is a feature of serrated lesions.

To investigate how BRAF$^{V600E}$ increases transcription of the cholesterol gene signature in human serrated lesions, we first applied the SCENIC method[34] to infer transcription factors and gene regulatory networks from scRNA-seq data of serrated lesions and normal cells. By doing so, we identified regulons that are specific to annotated cell types based on RSS[35] (Supplementary Fig. 15). Next, we employed iRegulon[36] to pinpoint which transcription factors active in SSCs were predicted to regulate the differentially expressed genes within the cholesterol gene signature (cut off ≥5 target genes). We singled out six transcription factors, namely CEBPB, CREB3, NR2F1, KLF16, SP6, and FOSL1, whose regulons are strongly enriched in SSCs (Fig. 6a) and which contribute towards the establishment of the cholesterol metabolism signature (Fig. 6b).

## Discussion
Ten percent of human CRCs, which originate from serrated lesions, harbor the V600E mutation in the BRAF oncogene[13]. This mutation is thought to contribute to tumorigenesis through the activation of the MEK/ERK arm of the MAPK pathway. Yet, despite the development of CRC being a decade-long process, the long-term impact of oncogene expression on the intestinal epithelium has not been investigated. Here, we assessed this issue using a genetically modified mouse model, the VillinCre$^{ER/0}$/Braf$^{V600E-LSL/+}$ (BVE), which enables the tamoxifen-inducible expression of mutant Braf$^{V600E}$ in the intestinal epithelium. We used a combination of transcriptomic and histological analyses to compare short-term and long-term impacts of Braf$^{V600E}$ expression. By doing so, we were able to identify changes in intestinal epithelial populations that were evident only after prolonged oncogene expression. Interestingly, despite failing to observe any loss of ISCs as reported by others[18,22], we did find downregulation of the ISC signature early after Braf$^{V600E}$ induction. However, ISC signature was recovered and downregulation reverted after 6 months. Although it is currently unclear what enables this adaptation in the BVE mice,

these changes could be explained by the well-established plasticity of the intestinal epithelial cells[37]. A striking alteration of Paneth cells was evident in 6-week and 6-month-induced animals. Paneth cells lost their localization at the base of the intestinal crypt and migrated towards the villi, where they displayed a rounded morphology and were similar to the intermediary cells expressing both goblet and Paneth cell markers described in ref. [22]. A similar phenotype was reported in other mouse models with intestinal expression of Braf$^{V600E}$, and a MAPK-dependent loss of Paneth cells was also described in models of mutant Kras[18,23,38]. Consistent with these observations, analysis of intestinal tissue after long-term Braf$^{V600E}$ activation revealed an impact on other secretory lineages. We observed changes in the numbers of both goblet cells and EECs. However, it should be noted that this reduction does not reflect the total number of cells. Indeed, since villi length was increased in mutant mice, cell numbers were normalized to villi length. Notwithstanding, the reduction in RNA levels of ChgA confirmed a general reduction in EECs. Whereas the increase in goblet cells probably reflects the mucinous nature of serrated neoplasia, the changes in EECs are particularly intriguing. Indeed, human BRAF$^{V600E}$ CRCs accumulate EEC progenitor cells that fail to complete their differentiation. These progenitor EECs secrete the TFF3 protein, a member of the Trefoil family, which supports CRC progression through activation of the PI3K/AKT pathway[39]. These data warrant additional research into the impact of BRAF mutations on the differentiation and activity of EECs in the context of serrated neoplasia.

Transcriptomic analysis through GSEA confirmed a negative enrichment of an intestinal differentiation signature after prolonged Braf$^{V600E}$ expression. This was likely driven by a significant downregulation of an enterocyte differentiation signature, which, to the best of our knowledge, has not been reported previously. The possible reduction in enterocytes was confirmed by reduced Alpl staining in intestinal tissue. Overall, our data suggest that the long-term impact of BRAF$^{V600E}$ is associated with a reduction in tissue differentiation, perhaps necessary to rebalance the initial decrease in stemness[18,22]. These data are in agreement with published evidence indicating that BRAF$^{V600E}$ suppresses hallmarks of intestinal differentiation, which can be restored upon pharmacological inhibition of the MAPK pathway[40].

The activation of Braf$^{V600E}$ led to a rapid and persistent enrichment in MAPK-dependent gene signatures associated with proliferation and CRC progression, including a fetal-like gene signature, a Myc signature, a G2/M signature and an E2F target gene signature. The latter two have been described as significant features of the BM2 molecular subtype of CRC[17]. We also observed the induction of a Wnt signaling-related signature, which was promptly downregulated by inhibition of the MAPK pathway. The association between the Wnt pathway and BRAF-driven CRC is still unclear. In mice, Rad and colleagues observed

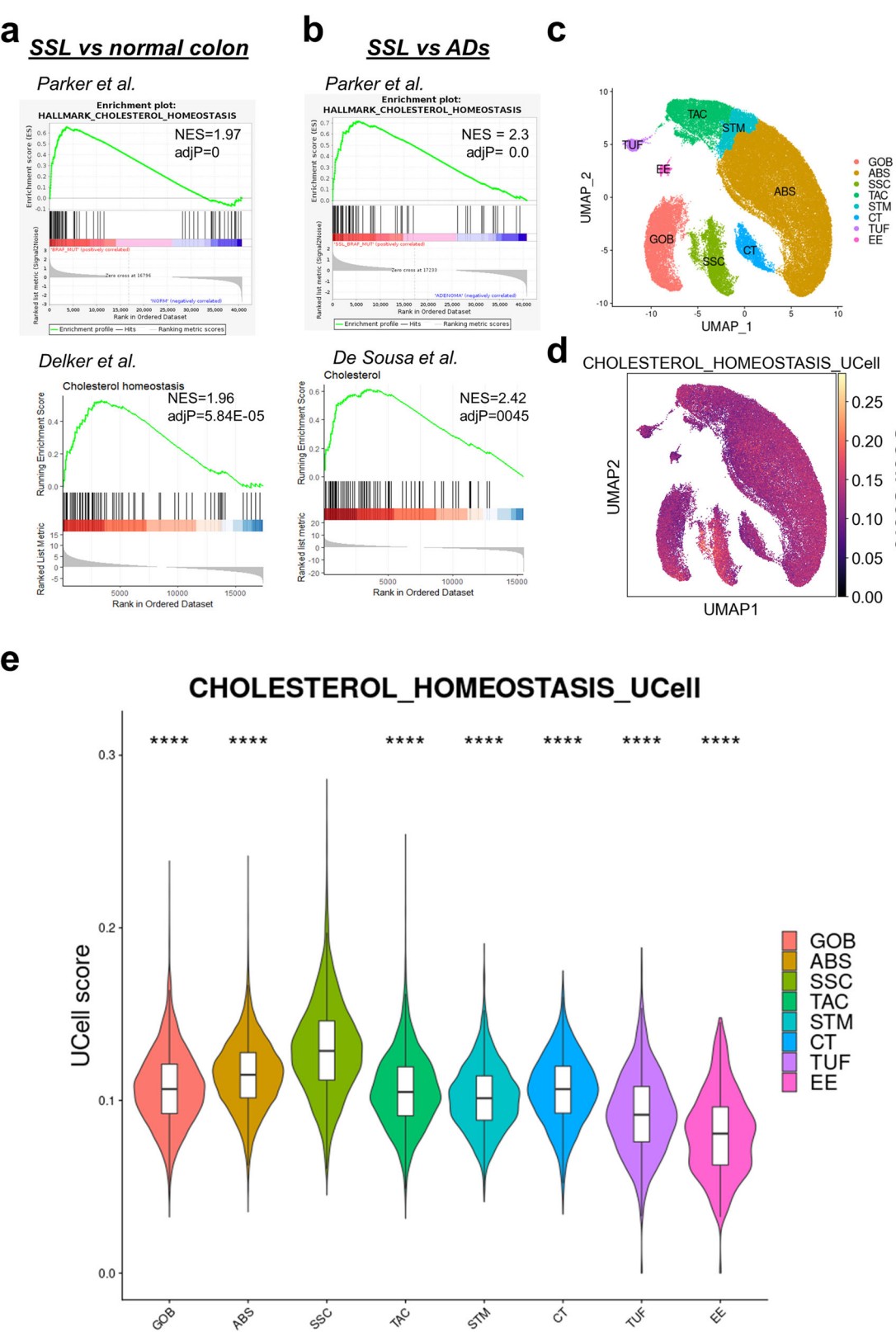

a progressive increase in Wnt activation during neoplasia progression[19], which was associated with nonsense or frameshift mutations in the *Apc* gene and activating mutations in the *Ctnnb1* gene, which encodes β-catenin. Mutations affecting Wnt-related genes have also been reported in human serrated CRCs. Truncating mutations of the Wnt negative regulator gene *RNF43* are common in MSI CRCs[41], and non-truncating mutations in the *APC* gene have also been detected[42]. However, the contribution of Wnt signaling to MSI CRCs remains controversial with some studies reporting lack of aberrant nuclear β-catenin localization[43], a surrogate marker for Wnt pathway activation, but others showing significant nuclear accumulation in more advanced

**Fig. 5 Gene signatures of cholesterol biosynthesis are enriched in transcriptomic datasets of human colorectal neoplasia. a, b** GSEA showing increased expression of the cholesterol gene signature in human serrated lesions (SSL) from three independent datasets comparing serrated neoplasia to normal matched control (**a**) or to adenomatous lesions (AD). NES normalized enrichment score. (**b**). **c** UMAP visualization of cells from serrated and normal datasets colored by annotated cell type. ABS absorptive cells, CT crypt top colonocytes, EE enteroendocrine cells, GOB goblet cells, SSC serrated specific cells, STM stem cells, TAC transient amplifying cells, TUF tuft cells. **d** UCell score distribution for the cholesterol gene signature shown in UMAP space. **e** Violin plot showing a comparison of UCell score distribution of a cholesterol gene signature in different cell populations. The boxes represent median and interquartile ranges. The whiskers show 95% confidence interval. The distribution of UCell score in SSC population was compared with the remaining cell lineages using the Wilcoxon statistical test. ****$P \leq 0.0001$.

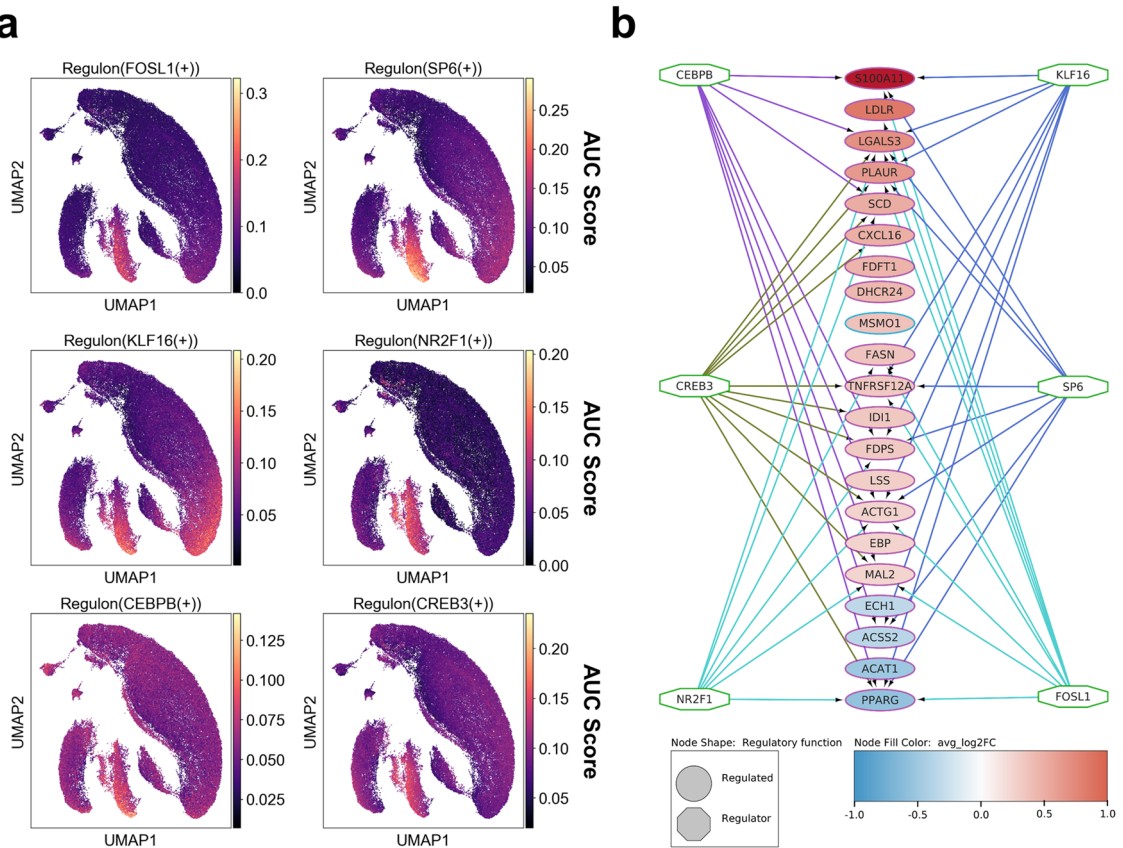

**Fig. 6 Analysis of regulons associated with the transcription of the cholesterol metabolism genes in a human single-cell transcriptomic dataset.
a** Regulon activity projected in UMAP space for the regulons associated with transcription factors (FOSL1, SP6, KLF16, NR2F1, CEBPB, CREB3) involved in the establishment of the cholesterol gene signature in the SSCs. **b** A network diagram generated with iRegulon in Cytoscape using the differentially expressed gene of the cholesterol biosynthesis signature in SSC identified by Seurat as an input. Circles indicate 21 significant genes, colored by avg_Log2FC. Octagons symbolize selected TFs (cut off >5 target genes) involved in the regulation of DEGs in the cholesterol biosynthesis signature. The edges represent connections between each of six transcription factors and their target genes and are colored based on the source of integration.

serrated lesions[42,44]. Notwithstanding, GSEA in BVE mice indicates that, despite the lack of neoplastic lesions, expression of *Braf*[V600E] triggers a transcriptional reprogramming congruent with tumorigenesis.

Metabolic rewiring is a common hallmark of cancer, including CRC[45–47] and we have recently, reported on the role of amino acid metabolism in CRC progression[48,49]. Here, we provide data that strongly support the establishment of a transcriptionally-driven metabolic reprogramming of cholesterol biosynthesis following activation of *Braf*[V600E]. We have also confirmed similar changes in human datasets of serrated lesions, suggesting that this metabolic adaptation is a feature of serrated neoplasia. Using bioinformatics tools SCENIC and iRegulon, we have finally identified the chief transcription factors that are responsible for the transcriptional rewiring: *CEBPB*, *CREB3*, *NR2F1*, *KLF16*, *SP6*, and *FOSL1*. The transcription factor CREB3 has been linked to regulation of cholesterol in response to the unfolded protein

response[50]. Similarly, the orphan nuclear receptors NR2F1 and 2 (also known as chicken ovalbumin upstream-promoter transcription factor, COUP-TFI and II) have been reported to regulate genes associated with cholesterol metabolism[51–53]. A role for FOSL1 in promoting cholangiosarcoma growth through regulation of HMG-CS1 along the mevalonate pathway has also been described[54]. Our data warrant further investigation to dissect the contribution of these transcription factors to the regulation of metabolic rewiring in *BRAF*-mutant CRCs and details of their regulation by MAPK. Importantly, statins have been suggested to possess cancer-preventive properties, but the association between the occurrence of adenoma or CRC and exposure to statins is controversial[28,55–57]. Very little is known about whether protection by statins depends on the specific molecular makeup of the tumor[58]. Notwithstanding, here, we show that treatment with atorvastatin prevents the establishment of crypt hyperplasia through induction of apoptosis in intestinal epithelial cells, but

without any evident impact on cell proliferation. However, it remains to be established how inhibition of cholesterol metabolism promotes apoptosis and how this relates to the presence of mutations in the *BRAF* oncogene. Moreover, our treatment study did not include non-induced mice and, therefore, we cannot formally rule out an effect of atorvastatin on normal intestinal epithelial cells. However, despite the inhibition of the mevalonate pathway displays some degree of unselective growth-suppressive function, tumor cells are more responsive than normal cells to pathway inhibition[59,60]. It is also noteworthy that atorvastatin treatment of *BRAF*-mutant mice reduces crypt cell numbers to levels comparable to WT animals, suggesting that statins likely revert the increased proliferation caused by the expression of mutant *BRAF*. The sensitivity of *BRAF*-mutant CRC cell lines to statin treatment has also been suggested to depend on BMP signaling through a functional *SMAD4* gene[61]. BMP signaling belongs to the TGF-β superfamily and mutations in the TGF-β pathway are common in *BRAF*-mutant CRCs. Whether this affects sensitivity to statins remains to be ascertained. Nonetheless, we observed a similar behavior in the response of *BRAF*-mutant CRC cells to atorvastatin in vitro: RKO cells with intact *SMAD4* showed sensitivity to statin treatment, whereas HT29 cell harboring homozygous mutations in *SMAD4* were resistant (Supplementary Fig. 16). These observations are in agreement with the evidence provided here that atorvastatin prevents crypt hyperplasia in the *Braf*-mutant intestine of BVE mice, which harbor a functional BMP pathway.

Overall, our data show that one of the most significant and consistent effects of *BRAF*^V600E^ expression in mouse tissue and human serrated lesions is the enrichment of a gene expression signature of cholesterol biosynthesis that contributes to the establishment of epithelial hyperplasia. These findings suggest that the incidence of *BRAF*-mutant CRC could be reduced using statins and warrant further targeted investigations into the specific statin sensitivity of this subtype of CRC.

Despite mouse models are suitable tools for recapitulating the development of CRC in vivo[62], they harbor defined genetic alterations within an inbred genetic background and carefully controlled, pathogen-free housing conditions, and thus do not fully recapitulate the complex genetic heterogeneity and micro-environmental complexity encountered in human tumors. In addition, a proper evaluation of the preventive activity of statins towards *BRAF*-mutant CRC would necessitate the assessment of tumor development in the *VillinCre*^ER^/*Braf*^V600E-LSL/+^ or other models of *BRAF*-driven colorectal carcinogenesis[11]. It should also be acknowledged that the influence of additional genetic alterations (such as mutation in *SMAD4*)[61,63] or/and environmental factors (such as inflammation or the microbiome)[64] could influence and perhaps overcome any protective effect of statins or alter tissue response to expression of the *BRAF* oncogene. A thorough analysis of statin anti-cancer activity in diverse experimental conditions is therefore warranted.

## Methods

**Cell lines and MTT assay.** HT29 CRC cells were grown in DMEM supplemented with 10% fetal calf serum (FCS) and Glutamax. RKO CRC cells were grown in MEM supplemented with 10% FCS and Glutamax. Cells were incubated at 37 °C and 5% $CO_2$. For MTT assay, 500–2000 cells were plated in 96-well plates. A 50 mM atorvastatin in DMSO stock solution was diluted in media (supplemented with dialyzed serum) to concentrations of 1, 5, and 10 μM. Three 96-well plates were run for each assay, each treated for either 24, 48, or 72 h. Cells were then incubated with MTT (2.5 mg/ml) at 37 °C 2 h. To dissolve the formazan produced by the metabolism of MTT, 50 μl of DMSO was added

to each well. The plates were then incubated for 30 min on a shaker before the optical density was read using a FluoStar OPTIMA plate reader (BMG-Labtech Ltd, UK).

**Mouse colony and genotyping.** All animal experiments were performed according to Home Office guidelines under project licenses (PPL) PC4E1710A and P7B8067BB. Experiments received ethical approval from the local ethics committee at the University of Leicester. All mice were C57BL6/j background aged between 8 weeks and 6 months. Unless otherwise stated in this section, figure legends and Supplementary Data 1, an equal number of males and females were used for experiments. Mice were housed in the pathogen-free Preclinical Research Facility (PRF) at the University of Leicester and were fed ad libitum with AIN93 diet (TestDiet, USA, Cat#5801-G), under a climate-controlled environment with 12 h day/night cycle. For transgene induction, 8–12-week-old double heterozygous *VillinCre*^ER/0^/*Braf*^V600E-LSL/+^ mice received intraperitoneal injections of 1 mg tamoxifen on 5 consecutive days at 24 h intervals, unless otherwise stated. Tamoxifen solution was prepared as a 10 mg/ml stock solution solubilized in corn oil. Control mice were *VillinCre*^ER/0^/*Braf*^+/+^ which received tamoxifen, or *VillinCre*^ER/0^/*Braf*^V600E-LSL/+^ treated with vehicle. After euthanasia, the small intestine was flushed with PBS and cut into six sections rolled to create a Swiss roll, and fixed in 4% [w/v] paraformaldehyde on a shaker overnight at room temperature. Fixed tissues were placed in embedding cassettes and submerged in 70% ethanol and processed by the Core Biotechnology Services (CBS) histology facility, the University of Leicester. Small pieces of tissue (~0.5 cm) were snap-frozen in liquid nitrogen for protein extraction or snap-frozen in RNAlater solution (Sigma) for RNA extraction.

DNA from mouse ear snips was extracted using the DNeasy Blood & Tissue Kit (QIAGEN) according to the manufacturer's instructions. PCR reaction mix for the *Braf* locus was: 10.5 μL of template DNA, OCP 125 (FWD primer) 1 μL (10 pmol), OCP 137 (REV-HET primer), 0.5 μL (5 pmol) OCP 143 (REV-WT primer) 0.5 μL (5 pmol), MyTaq Red (2×) (Meridian Bioscience BIO-25043) 12.5 μL. The PCR program was: 95 °C for 2 min, 30 cycles of 94 °C for 15 s, 60 °C for 15 s, 72 °C for 15 s; 5 min at 72 °C. For Cre^ER^ genotyping the PCR mix was: 10.5 μL of template DNA, OCP 361 (FWD primer) 1 μL (10 pmol), OCP 362 (REV primer) 1 μL (10 pmol), MyTaq Red (2×) (Meridian Bioscience BIO-25043) 12.5 μL. The PCR program was: 95 °C for 4 min, 30 cycles of 94 °C for 20 s, 57 °C for 20 s, 72 °C for 25 s; 5 min at 72 °C. Primer (Sigma) sequences are listed in Supplementary Table 1. The PCR products were visualized using 1.8% agarose gel with ethidium bromide (0.5 μg/mL) and imaged using the Syngene G:BOX Chemi XR5. Gels were examined for the presence of PCR bands at 140 bp (*Braf*^LSL-V600E^ allele), 466 bp (wild-type *Braf* allele), and 300 bp (*Villin-CreERT* allele).

**Statin treatment in vivo.** Thirteen-week-old female *VillinCre*^ER/0^/*Braf*^V600E-LSL/+^ mice received either 10 mg/kg atorvastatin by oral gavage or 5% DMSO/PBS vehicle control daily for 12 days. A 20 mg/mL atorvastatin (Generon) stock in DMSO was freshly diluted to a 1 mg/mL working solution in PBS. Days 1–7 of the study were a daily pre-treatment period of atorvastatin/vehicle. Days 8–10 consisted of both atorvastatin/vehicle gavage and intraperitoneal tamoxifen injection to induce *Braf*^V600E^ expression. On the final day of the study at 3 days post *Braf*^V600E^ induction, mice received intraperitoneal injections of 200 μL of 20 mM BrdU solution (Sigma) 3 h before tissue harvesting.

**Histology.** Tissue sections (4 μm), and hematoxylin and eosin (H&E) staining were performed by the Histology Facility, Core

Biotechnological Services, University of Leicester. Immunohistochemistry was performed using the Novolink Polymer Detection Systems kit (Leica Biosystems), according to the manufacturer's instructions. SuperFrost Plus™ Adhesion slides (Fisher Scientific) containing FFPE tissue sections were incubated at 65 °C for 30 min. Tissues were dewaxed and rehydrated through 3-min serial immersions in xylene, 99% industrial methylated spirit (IMS), and 95% IMS twice each, followed by 5 min in running tap water. Following antigen retrieval (Supplementary Table 2), slides were incubated for 5 min in a peroxidase block, washed twice in PBS, incubated for 5 min in the protein block solution and washed twice in PBS. Tissues were incubated with primary antibodies (Supplementary Table 2) (3% BSA/PBS) overnight at 4 °C. After incubation, sections were incubated in Polymer solution for 30 min, washed in PBS, and incubated in DAB solution for 5 min. Tissues were counterstained in Mayer's hematoxylin for 5 min and subsequently washed in tap water. Tissues were dehydrated through 3-min serial immersions in 95% IMS twice, 99% IMS twice, xylene twice. Slides were then mounted onto glass coverslips using DPX (Sigma Aldrich). Whole slide images were acquired using the Nanozoomer XR digital slide scanner (Hamamatsu Photonics) and visualized using NDP.view2 software. For immunofluorescence (IF), antigen-retrieved tissues were blocked in 5% BSA/0.3% Triton X-100 in PBS for 30 min, washed with 1% BSA/0.3% Triton X-100 in PBS twice for 10 min each and incubated with SP-1 Chromogranin A (ChgA) primary antibody (ImmunoStar) overnight at 4 °C. Slides were washed the next day in wash buffer solution twice for 10 min each. Alexa Fluor 568 secondary antibody (Invitrogen) was incubated for 1 h at RT in darkness. Slides were washed and sealed with coverslips using ProLong™ Glass Antifade Mountant containing DAPI (Invitrogen™). IF images were captured using the Vectra® Polaris™ imaging system (Akoya Biosciences), and images were viewed using Phenochart software v1.1.0 (Akoya Biosciences).

Alcian blue staining for visualization of goblet cells was performed using the Alcian Blue (pH 2.5) Stain Kit (Vector Laboratories). Rehydrated slides were heated at 65 °C and incubated in 3% acetic acid solution for 3 min, followed by incubation with Alcian Blue solution (pH 2.5) for 30 min at RT. Sections were rinsed in 3% acetic acid solution, running tap water, and distilled water. Sections were then counterstained with Enhanced Nuclear Fast Red (Vector Laboratories) for 5 min, rinsed in running tap water and distilled water, and dehydrated for mounting.

Intestinal alkaline phosphatase staining for the detection of enterocytes was performed using Vector Red alkaline phosphatase staining kit (Vector Laboratories). Rehydrated slides were incubated in substrate working solution for 30 min and washed in running water for 5 min. Tissues were counterstained in Mayer's hematoxylin (Leica Biosystems), dehydrated, and mounted in DPX.

**Electron microscopy (EM)**. All EM reagents were from Agar Scientific unless otherwise stated. For Transmission, Electron Microscope material was fixed overnight at 4 °C in 2.5% glutaraldehyde and 4% paraformaldehyde in 0.1 M cacodylate buffer, pH 7.2, and washed in 0.1 M cacodylate buffer. After the secondary fixation in 1% osmium tetroxide and 1.5% potassium ferricyanide (Merck Life Science) at RT, samples were treated with 1% tannic acid (VWR) dehydrated in ethanol series followed by propylene oxide (Merck Life Science) and embedded in Epon 812 (TAAB Laboratory Equipment Ltd). Samples were sectioned to 70-nm thick using a Reichert Ultracut E ultramicrotome, collected onto copper mesh grids, and stained for 5 min in lead citrate. Samples were viewed on a JEOL JEM-1400 TEM with an accelerating voltage of 120 kV. Digital images were collected with an EMSIS Xarosa digital camera with Radius software.

**Western blotting**. Tissues were homogenized in 400 μl of ice-cold RIPA buffer, containing protease inhibitor cocktail, and phosphatase inhibitor cocktail (Roche). The homogenized solution was kept on ice for 10 min with vortexing every minute. Samples were then centrifuged for 10 min at 12,000 rpm, at 4 °C. Supernatants were collected and protein concentration estimated with Lowry assay. Protein lysates were diluted to 1 mg/ml with SDS loading buffer (0.0625 M Tris-HCl pH 6.8, 2% SDS, 0.4 ml of 0.025% Bromophenol blue, 20% glycerol, 5% β-mercaptoethanol) and incubated at 95 °C for 10 min. In total, 15 μl of each sample was then run on acrylamide gels (Miniprotean III cell, BioRad) with pre-stained SDS-PAGE protein markers (All Blue Precision Plus Protein Standards, BioRad). Samples were then transferred to nitrocellulose membranes (Syngene) using a semi-dry transfer blot (BioRad) as per the manufacturer's instructions. For p-ERK1/2 (Rabbit, Cell Signaling, 9101 S), membranes were blocked in 5% BSA (Sigma Aldrich, A3059) in Tris-Buffered Saline, 0.1% Tween (TBST) for 90 min followed by incubation with primary antibody diluted 1:500 in 5% BSA in TBST overnight at 4 °C. For ERK2 (Mouse, Santa Cruz, sc-1647) membranes were blocked in 5% milk in TBST and incubated with the primary antibody diluted 1:1000 in 5% milk in TBST overnight at 4 °C. Membranes were washed in TBST, three times for 10 min each, and then incubated with secondary antibody (1:5000) at room temperature for 1 h (anti-Rabbit HRP Sigma A6154, anti-Mouse HRP Sigma A4416) followed by TBST washes three times. Bands were visualized using SuperSignal West Pico Chemiluminescent substrate kit (Thermo Fisher Scientific, 34580) and photographic film (Fuji).

**RNA extraction**. RNA was extracted from intestinal tissue on a Promega Maxwell 16 using the Maxwell 16 LEV simplyRNA Cells Kit (Reference AS1270). RNA quality and quantity was assessed using the 2100 Bioanalyzer Instrument (Agilent). The same RNA samples were used for RNA sequencing and Real-time PCR at 6 months.

**Transcriptomic analysis**
*RNA sequencing (RNA-seq)*. For RNA-seq, four *VillinCre^{ER/0}/Braf^{V600E-LSL/+}* 8–12-week-old mice were induced with tamoxifen and four were treated with vehicle to be left uninduced and used as controls to identify differentially expressed (DE) genes. Each experimental group contained two male and two female animals. Mice received intraperitoneal injections of 2 mg tamoxifen on 2 consecutive days. Intestinal tissue was harvested 6 months post induction. Indexed RNA libraries were prepared using the NEBNext® Single Cell/Low Input RNA Library Prep Kit for Illumina (NEB ref # E6420S) according to manufacturer's standard protocol and sequenced by 75 bp-paired-end sequencing on Illumina Novaseq, with the aim to obtain approximately 25 million reads per sample.

*Microarrays*. For microarrays, 12 mice 8–12 weeks old were split into two groups of six with an equal number of males and females. Six mice were *VillinCre^{ER/0}* controls, and six were *VillinCre^{ER/0}/Braf^{V600E-LSL/+}*. Both groups received 3 consecutive daily doses of 2 mg tamoxifen, and tissue was harvested 3 days after the last injection. Animals subjected to inhibition of MEK kinase with PD184352 treatment received two injections of 100 μl tamoxifen (10 mg/ml) on 2 consecutive days, followed by three injections of inhibitor/vehicle on 3 consecutive days. PD184352

(Selleckckem) was prepared in 10% DMSO (Sigma), 10% Cremophor EL (Calbiochem) in water. RNA was extracted as described above, and transcriptome profiling was conducted using the SurePrint G3 Mouse Gene Expression v2 8x60K Microarray Kit (Agilent, #G4852B), according to the manufacturer's instructions.

**Real-time PCR (rt-qPCR).** Reverse transcription of RNA was performed using the High-Capacity RNA-to-cDNA™ Kit (Applied Biosystems™) with a random primer approach. The 2X master mix was prepared according to the manufacturer's instructions, and 2 μg of purified RNA was added per 20 μL reaction. Reactions with and without the reverse transcriptase enzyme were performed in tandem, and samples without the enzyme were used as negative controls in subsequent rt-qPCR assays. The reverse transcription reaction was performed using the program: 25 °C for 10 min, 37 °C for 120 min, 85 °C for 5 min.

TaqMan® Gene Expression Assays (Fisher Scientific) were used for rt-qPCR reactions (Supplementary Table 3), using the StepOnePlus™ rt-qPCR System (Fisher Scientific) equipped with StepOnePlus™ Software v2.3. The thermocycler program was: 50 °C for 2 min, 95 °C for 20 s, then 40 cycles of: 95 °C for 1 s, 60 °C for 20 s. Each PCR reaction contained 1 μL of template DNA/control/water, 5 μL of 2× TaqMan Fast Advanced Master Mix (Applied Biosystems) and a TaqMan Gene Expression Assay (Thermo Scientific) containing 900 nM forward and reverse primers and 250 nM probe and 10 μL with nuclease-free water. Each sample was run in technical triplicates. rt-qPCR data were then analyzed using the ΔΔCt method. Ct values were normalized to endogenous control (EC) genes. We employed the NormFinder Excel Add-In, v0.953[65] to assess four candidate genes for use as endogenous controls: glucuronidase beta (*Gusb*), beta-2-microglobulin (*B2m*), beta-actin (*Actb*), and TATA-box binding protein (*Tbp*). The top three genes (*Gusb*, *B2m*, *Tbp*) with the most stable expression across experimental groups were selected as EC genes for the normalization of RT-qPCR assays in accordance with the MIQE guidelines[66].

**Bioinformatics analysis.** Bioinformatics analysis was performed using the statistical computing software R version 4.0.5 and the integrated development environment program RStudio version 1.3.1093. For RNA sequencing, FASTQ files sequencing quality was assessed using FastQC (v0.11.9). Sequencing adapter content was trimmed using FastP (v0.20.0)[67]. The pre-processed reads were then aligned to the mouse GRCm38/mm10 reference genome using Spliced Transcripts Alignment to a Reference (STAR) (v2.7.3a)[68]. Gene mapping and quantification were performed using DESeq2 (v1.34.0)[69]. For microarray data, the R Bioconductor package Linear Models for Microarray Data (Limma) (v3.50.1) was used to read raw text files from single-channel Agilent RNA microarrays and perform differential expression analysis.

**Gene set enrichment analysis (GSEA).** Agilent gene identifiers (IDs) from microarray analysis and RefSeq gene IDs from RNA sequencing were converted into Entrez IDs using BioMart (v2.50.3)[70] to ensure gene IDs were compatible with GSEA functions and The DE genes were ranked using Wald statistic. The R Bioconductor package clusterProfiler[71] was used to perform GSEA and statistical testing using Hallmark gene signatures from the Molecular Signatures Database (MSigDB)[72]. The *P* values were adjusted for multiple-hypothesis testing using the Benjamini–Hochberg method. Enrichment plots were generated using R Bioconductor package enrich plot (v1.14.2).

**Analysis of publicly available datasets.** Publicly available datasets (GSE106330[18], GSE168478[11], E-MTAB-6951[32], GSE45270[30], syn26720761 and TCGA-COAD[8]), were obtained from Gene Expression Omnibus (GEO) (https://www.ncbi.nlm.nih.gov/geo/), ArrayExpress (https://www.ebi.ac.uk/arrayexpress/), synapse (https://www.synapse.org/) and The Cancer Genome Atlas (TCGA) (https://www.cancer.gov/tcga). For GSE106330, normalized gene expression matrix was retrieved from GEO. For GSE168478, matrix of raw counts was obtained and further normalized using DESeq2 (v1.36.0) R package. Fastq files were downloaded from ArrayExpress for E-MTAB-6951. Reads, after quality control with fastQC (v0.11.5), were aligned to the human reference genome GRCh38 and quantified using STAR (v2.7.9a) with the following parameters: --quantMode GeneCounts, --outFilterScoreMinOverLread 0.33, --outFilterMatchNmin 40, --outFilterMatchNminOverLread 0.33. Count files were loaded into R and normalized using Deseq2 package. Expression CEL files were downloaded from GSE45270 and loaded into R using read.celfiles() function from affy R package (v1.74.0)[73]. Background correction normalization and expression estimation was done with rma() function. For syn26720761, processed TPM matrix was retrieved from synapse. For TCGA-COAD, the samples were downloaded in fastq format using GDC data transfer tool gdc client (v1.6.1)[74]. The quality of fastq files were analyzed with fastQC and samples, which failed quality control were removed. Universal Unique Identifier (UUID) of remaining samples was used to access TCGA-COAD STAR count matrices with TCGAbiolinks R package (v2.24.3)[75] and normalized with DESeq2 package. In total, 41 samples from normal tissue and 271 CRC samples having BRAF mutation status information (35 BRAF$^{V600E}$; 236 WT) were analyzed.

GSEA of analyzed public datasets was performed with GSEA software (v4.2.2)[76]. All matrixes, generated as described above, were saved in GCT format. The categorical class file format (cls) was created for each dataset to define phenotype labels. GSEA was performed using default settings except for the Permutation type set to gene_set.

**Single-cell RNA sequencing (scRNA-seq) data analysis**

*scRNA-seq, data subset, and integration.* Publicly available, scRNA-seq QC-filtered (Level 4) discovery (DIS) and validation (VAL) epithelial datasets were downloaded from HTAN data portal: https://data.humantumoratlas.org/ (HTAN Vanderbilt)[6]. To subset cells of interest, datasets were loaded into Python (v3.9.1) as AnnData objects (v0.7.6)[77]. Both datasets contained metadata attributes, such as polyp type and assigned cell type. This information was used to subset cells annotated as sessile serrated lesion and normal colonic biopsy with the *isin* function of the pandas (v1.3.1) package (Reback et al.[78]). In total, 64,382 normal cells (VAL: 34,008, DIS: 30,374) and 12,986 serrated cells (VAL: 6892, DIS: 6094) were subsetted for downstream analysis. Subsets of cells were loaded into R (v3.6.1) to perform integration with Seurat (v3.2.3) R package[79]. Prior to integration, datasets were independently log-normalized with the *NormalisationData* function. 2000 features being highly variable for each dataset and across datasets were identified using *FindVariableFeatures* and *SelectingIntegrationFeatures* functions, respectively. A *FindIntegrationAnchors* function with default settings was implemented to perform the unsupervised identification of cells that were used to integrate DIS and VAL datasets together by applying *IntegrateData* function. Further bioinformatics analysis was performed in Seurat (v4.1.1)[80].

*scRNA-seq, dimensional reduction, and clustering.* Principal component analysis (PCA) was performed on normalized and scaled data by implementing *RunPCA* function with a feature

argument set to use variable features. The first 10 dimensions of the PCA were used to calculate the neighbors with *FindNeighbors* function. Finally, the graph was segmented into clusters using *FindCluster* function with a resolution adjusted to 0.5. Uniform Manifold Approximation and Projection (UMAP), a non-linear dimensional reduction method, was selected to explore an integrated dataset by implementing *RunUMAP* function with a number of dimensions specified as 10. *DimPlot* function was further used to visualize cells in a low-dimensional UMAP space.

*scRNA-seq, cell-type annotation.* The original cell-type annotation, which was included in the metadata, was projected in the UMAP space with *DimPlot* function and used to rename 39 identified clusters by implementing *RenameIdents* function. In total, eight cell types were annotated within an analyzed subset of serrated and normal cells.

*scRNA-seq, gene signature scores, UCell.* UCell (v1.3.1) R package[33] was applied to evaluate cholesterol homeostasis signature distribution in serrated lesion and normal dataset, as well as in normalized and scaled original DIS and VAL datasets. List of genes involved in cholesterol homeostasis were retrieved from MsigDB[72]. UCell score for cholesterol signature was calculated by implementing *AddModuleScore_UCell* function on Seurat object. For serrated lesion and normal dataset, obtained cholesterol signature distribution in different cell types was visualized on UMAP space and a violin plot by implementing *FeaturePlot and VlnPlot* functions, respectively. For original DIS and VAL datasets, violin plots were used to compare cholesterol signature distribution in SSC and adenoma-specific (ASC) cell populations. For statistical analysis, mean comparison p values were added to violin plots with *stat_compare_means* function from ggpubr (v0.4.0) R package[81]. Means were compared using the nonparametric Wilcoxon rank sum test with SSC cluster as a reference group.

*scRNA-seq, differential expression testing.* The differential testing of expression of cholesterol biosynthesis genes between SSC and remaining cell populations was performed in Seurat. *FindMarkers* function was run with the "ident.1" argument defining SSC and "feature" argument specifying a list of cholesterol genes (74 genes) to test. As a result, 21 genes of cholesterol signature were identified as up- (17 genes) and downregulated (4 genes) in SSC population. In addition, the average expression of genes upregulated in SSC was visualized across all cell types with *DotPlot* function.

*scRNA-seq, gene regulatory network.* Single-Cell rEgulatory Network Inference and Clustering (SCENIC) method was applied to infer gene regulatory network in different cell types of serrated lesion and normal dataset. The required input files included scRNA-seq expression matrix, which was generated from Seurat object using build_loom function of ScopeLoomR R package. List of human transcription factors (1839 genes) and three hg19 gene-motifs ranking databases comparing ten species in .feather format were downloaded from https://resources.aertslab.org/cistarget/. First, the candidate regulatory modules of transcription factors and target genes were inferred by using the GRNBoost2 method (arboreto, v0.1.6). Next, RcisTarget was implemented to remove indirect targets from these modules based on motif discovery. Finally, the activity of predicted regulons was quantified at the cellular resolution by implementing the AUCell algorithm. Further, resulting regulon activity enrichment matrix was jointly analyzed with count-based matrix by using Scanpy (v1.9.1) python package[77]. To identify regulons that are specific to the SSC cluster we computed the regulon specificity score (RSS)[35]. The RSS was calculated for each cell type with

*regulon_specificity_scores* function of pySCENIC and the top 30 regulons for each cell type were shown on plot. Regulon activity of selected regulons was visualized on UMAP transcriptomic space with *sc.pl.umap* function of scanpy package.

*scRNA-seq, regulation detection of deregulated cholesterol genes.* A Seurat's table of 21 deregulated cholesterol genes in SSC was imported to Cytoscape (v3.9.1)[82], where iRegulon (v1.3)[36] was run with default settings to predict transcription factors potentially regulating the expression of these genes. Transcription factors that were shown previously to be specific for SSC were selected from the combination of motifs and tracks output (cut off >5 target genes). Generated regulatory network was visualized in Cytoscape with target genes node colored by avg_log2FC.

**Statistics and reproducibility.** Statistical significance testing was performed using GraphPad Prism 7 software. Comparisons between the two groups were performed using the two-tailed, unpaired Student's *t* test. Statistical analysis was performed experiments with more than three biological replicates. The number of replicates are reported in the figure legend for each experiment. Comparisons between more than two groups were carried on using ANOVA with appropriate post hoc analysis for multiple comparison (Tukey, Dunnet). Distribution data for intestinal crypts were analyzed using the Kolmogorov–Smirnov test. Attempts to reproduce the data were successful.

**Reporting summary.** Further information on research design is available in the Nature Portfolio Reporting Summary linked to this article.

### Data availability

Uncropped gel images for Supplementary Fig. 10b are reported in Supplementary Fig. 17. The transcriptomic data are available on GEO through accession numbers GSE236515 (RNA-seq data) and GSE234372 (microarray data). Any other data are available from the corresponding author upon reasonable request. Source data for generating graphs in Fig. 1b–d; Fig. 2e, f; Fig. 4c, e, f; Fig. 5e are reported in Supplementary Data 1. Source data for generating the graph in Fig. 4g are reported in Supplementary Data 2.

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

## Acknowledgements

This work was supported by the Cancer Prevention Research Trust (CPRT), a CRUK program grant (A13083) and a NIHR/CRUK-funded ECMC grant [C325/A15575]. S.W. was funded by a PhD fellowship from the CPRT (grant code TM60003-CPRT), I.G. was funded by a grant from the CPRT (grant code RM60G0665-CPRT). P.R. was funded by a grant from the CPRT (grant code RM60G0817). C.T. was supported by a Wellcome Trust Research Career Re-entry Fellowship (210911/Z/18/Z, 'Exploiting transcription of repetitive DNA to study early events in colorectal cancer'). R.P.G. was supported through a Development Grant to A.R. and C.T. from the charity Hope Against Cancer (https://www.hopeagainstcancer.org.uk/, grant code RM60G0751). We thank Dr. Sara Galavotti and Professor Andreas Gescher for their critical reading of the manuscript. The authors acknowledge the help from the staff of the Division of Biomedical Services, Preclinical Research Facility, University of Leicester, for technical support and the care of experimental animals. We thank the Advanced Imaging Facility (RRID:SCR_020967) and the University of Leicester Core Biotechnology Services Electron Microscopy Facility at the University of Leicester for their support.

## Author contributions

A.R. and C.P. designed the experiments. P.R., R.P.G., and M.M. performed bioinformatics analysis, S.W., P.F., H.C., I.G., S.S.U., C.A., L.S., C.G., E.P., H.J., F.H., and S.G. performed the experiments. N.B.S. supported microarray and RNA-seq experiments. N.S.A. and A.S-I. performed electron microscopy. A.R., C.P., K.B., C.T., and R.F. analyzed the data. A.R. and P.R. wrote the manuscript. All authors reviewed the manuscript.

## Competing interests

The authors declare no competing interests.
