## [Peer Review File · Communications Biology]

BRAFV600E-mutated serrated colorectal neoplasia drives transcriptional activation of cholesterol metabolism.Reviewers' comments:

Reviewer #1 (Remarks to the Author):

Rzasa, Whelan, and Farahmand, et al. utilized a mouse model of mutated BRAF-driven tumorigenesis to identify early and late transcriptomic perturbations of intestinal homeostasis.

At a histologic level, upon induction of BRAF V600E, the authors' work reveals evidence of early hyperplasia and subsequent shift towards decreased differentiation relative to stemness. Transcriptomic analysis of intestinal tissue showed a BM2-like gene expression signature, in addition to evidence of dysregulation of cholesterol metabolism. The findings were correlated with bulk and single-cell gene expression patterns in human sessile serrated lesions, demonstrating upregulation of a cholesterol biosynthesis signature relative to normal human colon tissue and traditional adenomatous lesions.

Overall, the work provides significant novel insights into the mechanisms of early BRAF-driven tumorigenesis. The analytical framework is appropriate and well described.

Major comments:

1. In vitro, the authors observed a significant decrease in crypt hyperplasia within braf-mutated mice treated with atorvastatin versus vehicle control. how do the results compare to WT mice controls? I.e., can the authors exclude that treatment with atorvastatin alone lead to significant change in the number of cells per crypt or proliferation?
2. Based on the findings, the authors speculate that statins may prevent BRAF-mutated CRC, perhaps conditional on a functioning BMP pathway. However, the functional relevance of the cholesterol biosynthesis adaptations was only investigated at 3-days post induction of mutant braf. To more directly test their hypothesis, did the authors observe significant differences in mouse survival and/or tumor burden in braf-mutated mice treated for longer durations of time with atorvastatin?

Reviewer #2 (Remarks to the Author):

This study highlights the importance of investigating the long-term effects of oncogene expression on tissue homeostasis and the potential of repurposing existing drugs such as statins to prevent the development of specific types of cancers.

Overall, the study provides novel insights into the long-term effects of BRAF mutations on the homeostasis of intestinal tissue, which may help in the development of new strategies for the prevention and treatment of colorectal cancers. There are several concerns regarding on this study:

1. While inducible mouse models provide a controlled system for investigating the effects of oncogene expression, they have limitations in terms of their ability to accurately reflect the complex interactions between different cell types and microenvironmental factors that occur in human tissues.
2. The study investigated the effects of short-term (3 days) and long-term (6 months) expression of BrafV600E in the intestinal tissue of mice. However, it is possible that longer-term expression of the oncogene or exposure to other environmental factors may lead to different results.
3. While the study found an increased expression of the cholesterol biosynthesis signature in human serrated lesions, it is unclear whether this signature is specific to BRAF-mutant colorectal cancers or if it is also present in other types of colorectal cancers or even in other cancers.
4. The study's bioinformatic analysis indicated that atorvastatin prevented the establishment of

intestinal crypt hyperplasia in BrafV600E-mutant mice, but it is important to validate this result in the actual mouse model through further experiments.

RE: “Rzasa, Whelan, Pooyeh Farahmand et al. Transcriptional activation of cholesterol metabolism in BRAFV600E-driven serrated colorectal neoplasia.”

We would like to thank both reviewers for the time they dedicated to read and evaluate our manuscript. We are pleased that they found that our findings could further our understanding of serrated neoplasia.

We hope they will find that our revised manuscript has addressed their main concerns.

Please note that some graphs have been modified to reflect the journal’s guidelines and figure legends updated accordingly.

POINT-BY POINT RESPONSES TO REVIEWER’S COMMENTS

Response to Reviewer #1:

Rzasa, Whelan, and Farahmand, et al. utilized a mouse model of mutated BRAF-driven tumorigenesis to identify early and late transcriptomic perturbations of intestinal homeostasis.

At a histologic level, upon induction of BRAF V600E, the authors' work reveals evidence of early hyperplasia and subsequent shift towards decreased differentiation relative to stemness. Transcriptomic analysis of intestinal tissue showed a BM2-like gene expression signature, in addition to evidence of dysregulation of cholesterol metabolism. The findings were correlated with bulk and single-cell gene expression patterns in human sessile serrated lesions, demonstrating upregulation of a cholesterol biosynthesis signature relative to normal human colon tissue and traditional adenomatous lesions. Overall, the work provides significant novel insights into the mechanisms of early BRAF-driven tumorigenesis. The analytical framework is appropriate and well described.

We are thankful to this reviewer for their positive evaluation of our work.

Major comments:

1. In vitro, the authors observed a significant decrease in crypt hyperplasia within braf-mutated mice treated with atorvastatin versus vehicle control. how do the results compare to WT mice controls? I.e., can the authors exclude that treatment with atorvastatin alone lead to significant change in the number of cells per crypt or proliferation?

RE: This is an excellent point! Unfortunately, wild-type control mice were not included in the treatment experiment, and we cannot provide this data in the current resubmission. Notwithstanding, we note that atorvastatin treatment of BRAF-mutant mice reduces crypt cell numbers to levels comparable to WT animals, suggesting that statins likely revert the increased proliferation caused by expression of mutant BRAF. Moreover, despite the inhibition of the mevalonate pathway displays some degree of unselective growth-suppressive function, tumour cells are more responsive than normal cells to pathway inhibition. Please see references:

1. Mo H, Elson CE. Studies of the isoprenoid-mediated inhibition of mevalonate synthesis applied to cancer chemotherapy and chemoprevention. *Exp Biol Med (Maywood)*. 2004 Jul;229(7):567-85.
2. Wong WWL, Dimitroulakos J, Minden MD, Penn LZ. HMG-CoA reductase inhibitors and the malignant cell: the statin family of drugs as triggers of tumor-specific apoptosis. *Leukemia*. 2002 Apr;16(4):508-19.

We have now discussed this limitation of the study quoting the above references. We added the following paragraph to our discussion section:

“Moreover, our treatment study did not include non-induced mice and, therefore, we cannot formally rule out an effect of atorvastatin on normal intestinal epithelial cells. However, despite the inhibition of the mevalonate pathway displays some degree of unselective growth-suppressive function, tumor cells are more responsive than normal cells to pathway inhibition^{59,60}. It is also noteworthy that atorvastatin treatment of BRAF-mutant mice reduces crypt cell numbers to levels comparable to WT animals, suggesting that statins likely revert the increased proliferation caused by expression of mutant BRAF”

2. Based on the findings, the authors speculate that statins may prevent BRAF-mutated CRC, perhaps conditional on a functioning BMP pathway. However, the functional relevance of the cholesterol biosynthesis adaptations was only investigated at 3-days post induction of mutant braf. To more directly test their hypothesis, did the authors observe significant differences in mouse survival and/or tumor burden in braf-mutated mice treated for longer durations of time with atorvastatin?

RE: We agree with this reviewer. Ideally, a long-term experiment should be performed to address the impact of statin treatment on cancer onset. However, because of the rate and incidence of tumour development in the VillinCre^{ER/0}/Braf^{V600E-LSL/+} (Supplemental Figure 1) a long-term experiment of over a year using a substantial number of animals would be required. Unfortunately, such an experiment would be financially challenging and incompatible with the timeline of this revision.

Response to Reviewer #2:

This study highlights the importance of investigating the long-term effects of oncogene expression on tissue homeostasis and the potential of repurposing existing drugs such as statins to prevent the development of specific types of cancers.

Overall, the study provides novel insights into the long-term effects of BRAF mutations on the homeostasis of intestinal tissue, which may help in the development of new strategies for the prevention and treatment of colorectal cancers. There are several concerns regarding on this study:

We thank this reviewer for their assessment of our paper and the insightful comments.

1. While inducible mouse models provide a controlled system for investigating the effects of oncogene expression, they have limitations in terms of their ability to accurately reflect the complex interactions between different cell types and microenvironmental factors that occur in human tissues.

RE: We agree with the reviewer’s sensible comment to take into consideration the limitations of our experimental models. We have acknowledged this by adding a statement at the end of the discussion section:

“Despite mouse models are suitable tools for recapitulating the development of CRC in vivo, they harbor defined genetic alterations within an inbred genetic background and carefully controlled, pathogen-free housing conditions, and thus do not recapitulate fully the complex genetic heterogeneity and microenvironmental complexity encountered in human tumors. In addition, a proper evaluation of the preventive activity of statins towards BRAF-mutant CRC would necessitate the assessment of tumor

development in the VillinCre^{ER/0}/Braf^{V600E-LSL/+} or other models of BRAF-driven colorectal carcinogenesis. It should also be acknowledged that the influence of additional genetic alterations (such as mutation in SMAD4) or/and environmental factors (such as inflammation or the microbiome) could influence and perhaps overcome any protective effect of statins or alter tissue response to expression of the BRAF oncogene. A thorough analysis of statin anti-cancer activity in diverse experimental conditions is therefore warranted.”

2. The study investigated the effects of short-term (3 days) and long-term (6 months) expression of BrafV600E in the intestinal tissue of mice. However, it is possible that longer-term expression of the oncogene or exposure to other environmental factors may lead to different results.

RE: Please see our response to the previous comment

3. While the study found an increased expression of the cholesterol biosynthesis signature in human serrated lesions, it is unclear whether this signature is specific to BRAF-mutant colorectal cancers or if it is also present in other types of colorectal cancers or even in other cancers.

To address this point, we have reanalysed the HTAN dataset in Figure 5E to include single cell data from traditional adenomas (harbouring no mutations in BRAF). The HTAN dataset (Chen B et al. Cell. 2021 Dec 22; 184(26): 6262–6280.e26) comprises 9 patients with SSLs, 8 of which harbour mutant BRAF. We could confirm that SSL enriched for mutant BRAF have increased expression of the cholesterol gene signature compared to traditional adenomas. We then confirmed enrichment in the cholesterol gene signature in human BRAF-mutant CRC in an independent bulk RNAseq dataset (Joanito et al. Nat Genet. 2022 Jul;54(7):963-975). Then, we analysed colon adenocarcinoma (COAD), thyroid cancer (THCA) and skin cutaneous melanoma (SKCM) from the TCGA cohorts, as they contained sufficient BRAF cases for analysis. Unfortunately, we did not identify sufficient BRAF mutations to perform such analysis in the pancreatic and lung cancer cohorts. The results from the TCGA analysis are reported in the table below. Only COAD and THCA showed a significant enrichment for the cholesterol signature in mutant BRAF tumours. The data on CRC are now presented in the revised Figure 5 and Supplemental Figure 13A. Since the focus of this manuscript is BRAF mutant CRC, we decided not to include results from other malignancies in the revised manuscript, unless this reviewer should advise otherwise.

TCGA dataset	No of samples		NES	Nominal p-value
	BRAFV600E	WT		
COAD	35	236	1.8	6E-4
THCA	283	205	1.49	0.008
SKCM (TP)	49	54	0.942	0.556
SKCM (TM)	150	217	-1.070	0.336
SKCM (TP&TM)	199	271	-1.023	0.421

*TP=primary tumour
TM= metastatic tumour
NES=normalised enrichment score*

4. The study's bioinformatic analysis indicated that atorvastatin prevented the establishment of intestinal crypt hyperplasia in BrafV600E-mutant mice, but it is important to validate this result in the actual mouse model through further experiments.

RE: Although we have shown that atorvastatin treatment restrains intestinal crypt hyperplasia through induction of apoptosis, we appreciate this reviewer's comment that additional work would be necessary, as there are many additional questions to answer: does BRAFV600E prime intestinal epithelial cells for apoptosis? How does cholesterol metabolism protect from apoptosis? Etc.. However, we hope that this reviewer appreciates that this work is beyond the scope of the current submission and will be addressed in future work from our team. We have however addressed this reviewer's comments in the discussion of the revised manuscript adding the following paragraph:

"Notwithstanding, here, we show that treatment with atorvastatin prevents the establishment of crypt hyperplasia through induction of apoptosis in intestinal epithelial cells, but without any evident impact on cell proliferation. However, it remains, to be established how inhibition of cholesterol metabolism promotes apoptosis and how this relates to the presence of mutations in the BRAF oncogene."

REVIEWERS' COMMENTS:

Reviewer #1 (Remarks to the Author):

Rzasa, Whelan, and Farahmand et al. provide a revised manuscript describing the impact of statin treatment in a mouse model of BRAF-driven colorectal tumorigenesis.

The authors have adequately addressed concerns and comments regarding their original submission.

Reviewer #2 (Remarks to the Author):

All my concerns have been well addressed!